# OpenReview forum: "ReSink: Stop Words to Improve Training-Free Referral Segmentation"
_TMLR — Under review for TMLR_

### Review · Reviewer_LkHm · 2026-05-28

**Summary Of Contributions:**

The paper proposes to add stop words as attention magnets for DiT-based referring segmentation.

While the previous work ConceptAttention (Helbling et al., 2025) introduces concept (object) tokens for zero-shot object localization, this work turns it around to study stop word tokens for background suppression.

Based on this idea, the paper introduces the RESINK method and demonstrates its state-of-the-art performance on zero-shot referring image/video segmentation.

Overall, the idea is interesting and the paper is generally written well.

**Additional Comments:**

The references need update. For example, ConceptAttention (Helbling et al., 2025) was published in ICML 2025.

In addition, a few referring segmentations works are not included for discussion/comparisons: [A] is a recent zero-shot referring segmentation method that directly relates to this work, while [B,C] are self-supervised methods that do not require additional ground truth for training, which is partially related.

[A] IteRPrimE: Zero-shot Referring Image Segmentation with Iterative Grad-CAM Refinement and Primary Word Emphasis, AAAI 2025

[B] Shatter and Gather: Learning Referring Image Segmentation with Text Supervision, ICCV 2023

[C] Referring Image Segmentation Using Text Supervision, ICCV 2023

**Audience:**

Yes

**Audience Explanation:**

The paper presents an interesting idea that leverages stop word tokens for zero-shot referring segmentation. The idea may generalize to other tasks.

**Broader Impact Concerns:**

The paper has discussed broader impact concerns.

**Claims And Evidence:**

Yes

**Claims Explanation:**

While the paper provides evidences to support its claims in general, there are a few pieces of evidence remain unclear.


(1) The method focuses on background suppression, while how it infers the referred objects from the properties/relations described by the text descriptions remain unclear. This also reflected by the visual results (e.g., Figure 5) where the model seems to locate the target objects based on the class/concept described in the text. To demonstrate it specifically tailored for referring segmentation, clarification on how it reasons from relation/property descriptions and corresponding visual results are essential. In addition, while the paper says stop words + colors yield best performance, I wonder if it may introduce noise for cases that rely on colors (e.g., pink/black) for referring (How do we determine the color tokens for augmentation?). Last, the experimental setting for Table 8 needs clarification. How does the paper apply ConceptAttention to referring segmentation task?


(2) First, a few statistical numbers need clarification, e.g., 77% (10%/10%) of GAS tokens correspond to stop words (color tokens/others); and 89% of cases where color-GAS tokens reassign mass to magnets. It is not clear how these numbers are obtained.
Second, some experimental settings require clarifications, e.g., the backbone model used in Figure 4/Table 1, and the experimental settings for Table 1.

(3) The backbone-agnostic claim needs justification. The method needs to collect features from different time-steps for different models. The paper may need to explain whether they are specifically tuned or if there is a robust criterion.

(4) I think the core ideas of CrossAttention and RESINK are not contradictory. Have the authors considered combining both together?

**Requested Changes:**

Please see the first three points in Section 2.

---

> ### Author Response · Authors · 2026-07-03
> **Official Comment by Authors (1/3)**
>
> We thank the reviewer for the constructive and detailed feedback, and for finding the idea interesting and the paper well written. Highlighted (green) passages mark the corresponding edits in the revised PDF.
>
> > (1a) How does the method reason over relations/attributes, not just the object category?
>
> We agree this deserves explicit treatment. ReSINK grounds the referral expressions through the emergent DiT attention:
>
> 1. The discriminative power here comes from cross-attention itself, powered by the semantic understanding already built into the underlying text encoder, not a separate reasoning step. A noun token on its own highlights every instance of that category in the scene, with varying strength, but other content tokens carry their own, only partly overlapping, spatial signal. Averaging across these tokens is what does the work: a region reinforced across several tokens survives the average, while a region supported only by the class token is suppressed. This is consistent with our finding in Sec. 4.3 that the T5 encoder is structure-aware, removing stop words costs it several points, while CLIP behaves like a bag-of-words; the compositional signal we exploit comes from the language model's own representation, not from our pipeline. Figures already in the submission show this directly. In Fig. 1, the per-token map for "largest" is what isolates the correct fish among several goldfish, not the class token "goldfish." In Fig. 14 ("a gray colored mountain goat"), without magnets the argmax lands on "mountain" instead of the goat; with magnets it correctly moves to "goat," the actual head noun. Fig. 9 (Reasoning over relations and attributes) shows this directly for referring expressions: for the relational expression "left guy" the prediction localizes the correct person among two, and for the attribute expression "girl in purple" it lands on the attributed instance, confirming the method resolves the full expression, not only the object class.
>
> 2. Additionally, we handle relational expressions like "left" or "right" the same way HybridGL (Liu & Li, 2025) and Ref-Diff (Ni et al., 2023) do: rather than expecting the core representation to encode position, both detect a small set of relational keywords and apply a separate positional bias only when one appears. This matches the well-documented finding that vision-language and text-to-image models struggle with these distinctions even when the objects themselves are correctly recognized [A1a; B1a]. Our noun-phrase / spatial-bias module (Appendix C) follows the same approach: relative relations compare candidate-region centroids, and absolute terms map to positional priors over the image grid, added on top of the attention signal rather than replacing it.
>
> This also holds at scale. A breakdown by expression type on RefCOCO val shows attribute expressions reaching the highest oIoU/mIoU (53.7/63.3) and relational expressions performing on par with plain ones, despite carrying more same-category distractors:
>
>  | subset (RefCOCO val, 3k expr)          | n    | oIoU | mIoU | PA   |
>   | -------------------------------------- | ---- | ---- | ---- | ---- |
>   | spatial (relational, e.g. "left guy")  | 1827 | 41.5 | 51.3 | 63.0 |
>   | plain                                  | 676  | 42.9 | 49.8 | 60.9 |
>   | attribute (e.g. "girl in purple")      | 497  | 53.7 | 63.3 | 74.2 |
>
> [A1a] Kamath et al., What's Up with VLMs? Spatial Reasoning, EMNLP 2023.
>
> [B1a] Gokhale et al., Benchmarking Spatial Relationships in T2I Generation, arXiv:2212.10015.
>
> > (1b) Do stop-words + color magnets add noise for color-referring expressions? How are augmentation colors chosen?
>
> The mechanism is designed precisely to avoid this failure mode:
>
> - **Appended magnets are distinct, later-filtered tokens.** Even if an appended color magnet and an in-expression color word share the same string, they produce separate attention contributions: the T5 text encoder encodes each occurrence in its sentence context, so the two tokens sit at different sequence positions and receive different contextual embeddings, yielding different cross-attention maps. We filter the appended magnets but keep the in-expression attribute token in $H_e$. The discriminative color cue is therefore retained.
> - **The purpose is the opposite of erasing color cues.** When a Global Attention Sink lands on a meaningful color token, the appended magnet provides a decoy destination so the **original** color token recovers specificity.
> - **Empirically the color magnets help, not hurt** (Tab. 5): adding color words to the magnet set improves 𝓙&𝓕 from 56.8 to 57.6, a net gain, not a loss.
> - **Choice of colors:** a small **fixed** auxiliary set ("pink", "blue", "green", "white"), applied identically across all datasets and models, with **no per-instance or per-dataset tuning** (Sec. 4, Implementation Details). Robustness to the magnet set is shown in Tab. 5 (different stop-word sets excluding colors yield near-identical results).

---

> ### Author Response · Authors · 2026-07-03
> **Official Comment by Authors (2/3)**
>
> > (1c) Clarify Table 8, how is ConceptAttention applied to referring segmentation?
>
> Clarified in Appendix A.2 (highlighted). The representation-space comparison (Ref-DAVIS17) uses the **same** ReSINK referring pipeline (same referring-expression tokens, same aggregation, same argmax→SAM2); the rows differ **only** in the representation used to build the heatmap:
>
> - **"Cross Attention"** (our default): per-token cross-attention maps.
> - **"Attention Output"** (ConceptAttention's space): image-patch attention *outputs* projected onto the referring-expression token embeddings.
>
> Crucially, we do not use CA's original protocol (which requires a pre-enumerated set of single-word scene concepts); we drive its representation space with the referring expression, so the comparison isolates the representation space alone. Cross-attention is better for referring: **57.6 vs 55.6 𝓙&𝓕** on Ref-DAVIS17. We also compare directly on image segmentation under the **same FLUX DiT backbone**, where ReSINK outperforms both ConceptAttention and Flux Cross-Attention: ImageNet-Seg mIoU **71.37 vs 71.04 vs 59.90**, PascalVOC mIoU **78.57 vs 76.45 vs 54.77**, isolating the algorithmic gain from the backbone.
>
>
>
> > (2) How are the statistics obtained, 77/10/10 GAS split, 89% color-GAS reassignment, and the attractor evidence?
>
> We clarify the full measurement protocol and add it to Sec. 3.4. All statistics are computed over N = 1000 referring expressions from RefCOCO testA.
>
> **GAS identification.** For each expression we compute, per text token, its layer-averaged text-to-text attention mass, and mark a token as a Global Attention Sink (GAS) if this mass exceeds **10× the mean across all tokens** (the criterion stated in Sec. 3.4; typically 1-3 GAS per sequence).
>
> **Token-type distribution (77 / 10 / 10).** Each identified GAS token is categorized as a stop word (the NLTK set, extended with the tokenizer symbols `.`, `,`, `_`), a color word, or another content word; aggregating over the 1000 expressions yields 77% stop words, 10% color words, 10% other content words. The color/content cases are the harmful ones, since a GAS on a meaningful token suppresses its discriminative signal, which motivates the redistribution mechanism.
>
> **Color-GAS reassignment (89%).** Restricting to expressions where a GAS falls on a color token, we append the color magnets and measure the fraction of cases in which the surplus attention mass moves onto the appended magnet (freeing the original color token): this occurs in ~89% of cases.
>
> **Attractor, not aggregator.** To substantiate that GAS carry no useful localization signal, we report (same N = 1000, GAS vs. normal prompt tokens):
>
>   | statistic | GAS (sink) | normal prompt token |
>   |---|---|---|
>   | background attention mass | **88.9%** (foreground 11.1%) | n/a |
>   | value-vector L2 norm | **95.5** | **126.3** |
>   | cosine similarity to image features | **$\approx$ -0.01** | n/a |
>   | attention entropy | **8.31** | 8.04 (max $\approx$ 8.32) |
>
> These show sinks absorb attention from irrelevant (background) regions, have **lower** value-norm (low output impact → attractor, not aggregator), do not aggregate image semantics (near-zero cosine), and are diffuse (near-maximal entropy). This protocol is now stated explicitly in Sec. 3.4 (highlighted).
>
> **Backbone / settings for Fig. 4 and Table 1.** Both are computed with **Mochi on Ref-DAVIS17** (the RVOS setting; 𝓙&𝓕). "Filtered Blocks = X%" excludes the earliest X% of transformer blocks from the attention aggregation; performance is unchanged up to 60% filtering. Added to the captions (highlighted).
>
>
>
> > (3) Backbone-agnostic claim: are the per-backbone timesteps tuned?
>
> Not tuned. The timesteps follow the well-established finding that **mid-trajectory** diffusion features carry the richest semantics; image (FLUX/SD3.5) and video (Mochi) backbones use different noise schedules, so the same mid-band maps to different absolute indices (520/640 for FLUX/SD3.5; 990 for Mochi). We use a **single fixed choice per backbone across all datasets**, with zero per-dataset tuning.
>
> We add a **timestep-sensitivity ablation** (RefCOCO val subset; new figure in the revised appendix) showing a flat plateau across the mid-band, oIoU stays within ~**40-43** (mIoU ~51-54) across t ≈ 440-700, matching the headline RefCOCO val (Table 2), confirming the choice is robust rather than tuned.

---

> ### Author Response · Authors · 2026-07-03
> **Official Comment by Authors (3/3)**
>
> > (4) ConceptAttention and ReSINK are not contradictory, have you considered combining them?
>
> We fully agree they are complementary, and our design reflects this. ConceptAttention contributes a *representation-space* idea (reading attention in the output/projection space); ReSINK contributes (i) attention-magnet redistribution and (ii) a referring-expression-driven setup that removes CA's requirement to pre-enumerate scene concepts. Our pipeline already evaluates CA's representation as one option (Ref-DAVIS17: cross-attention **57.6** vs attention-output **55.6** 𝓙&𝓕), and on the same FLUX backbone ReSINK outperforms CA on image segmentation (e.g. ImageNet-Seg mIoU 71.37 vs 71.04). Because the magnet mechanism manipulates *which* tokens absorb attention under the softmax constraint, independent of whether the final heatmap is read from cross-attention or the output space, "combining" the two simply amounts to applying ReSINK's magnets on top of CA's representation, a compatible extension our framework already supports. We make this complementarity explicit in Sec. 3.2 / Appendix A.2.
>
>
>
> > References & related work
>
> - **ConceptAttention** citation updated to **ICML 2025**.
> - Added to Related Work:
>   - **[A] IteRPrimE** (AAAI 2025), zero-shot RIS via iterative Grad-CAM refinement and primary-word emphasis; closely related. We discuss the distinction (we exploit emergent DiT attention sinks + redistribution vs. iterative CAM refinement).
>   - **[B] Shatter-and-Gather** (ICCV 2023) and **[C] Text-supervised RIS** (ICCV 2023), text/weakly-supervised RIS; discussed as related (they avoid mask GT but still train, whereas ReSINK is fully training-free).
> - Bibliography metadata filled for all four.

---

### Review · Reviewer_wg1c · 2026-06-08

**Summary Of Contributions:**

The paper introduces ReSINK, a training-free framework for zero-shot referring image and video segmentation. The key contributions are:
1.	Identification of Global Attention Sinks (GAS) in Diffusion Transformers (DiTs): The paper shows that certain tokens act as "attention sinks," absorbing disproportionately high attention across both text and image tokens in deeper layers, while carrying no useful semantic signal for grounding.
2.	Attention Redistribution via Stop Words: Based on the GAS phenomenon, the paper proposes a simple yet effective mechanism: appending additional stop words to the referring expression. These act as "attention magnets" that absorb surplus background attention and can redirect harmful GAS behavior away from meaningful tokens.
3.	A Unified, Training-Free Pipeline: The proposed ReSINK method combines cross-attention map extraction, filtering of stop words, aggregation of the remaining maps, and a final SAM-based segmentation step. This pipeline works for both images and videos without any fine-tuning or architectural modifications.
4.	State-of-the-Art Performance: The paper demonstrates that ReSINK achieves new state-of-the-art or highly competitive results on multiple zero-shot referring image and video segmentation benchmarks, outperforming prior training-free methods.
Key Strengths:
1.	Novelty & Insight: The transfer of the "attention sink" concept from LLMs to DiTs for a downstream vision-language task is insightful and clever. Using stop words as controllable "magnets" is a simple, elegant solution.
2.	Simplicity & Generality: The method is remarkably simple, training-free, and backbone-agnostic, making it highly practical and easy to reproduce.
3.	Strong Results: The quantitative results are compelling, showing consistent gains over prior zero-shot methods and even closing the gap with some supervised approaches on several benchmarks.
4.	Thorough Analysis: The paper provides good ablations and analysis, which strongly support the claims.
Key Weaknesses:
1.	Dependence on Additional Models: While training-free, the pipeline depends on several large foundation models. The inference efficiency is good, but the overall system complexity is non-trivial.
2.	Temporal Limitation: The video segmentation method ignores temporal aspects of expressions and always localizes in the first frame. This is a significant simplification for RVOS.
3.	Point-based Segmentation Limitation: Using a single argmax point to prompt SAM can lead to under-segmentation, which is acknowledged but not solved.

**Audience:**

Yes

**Audience Explanation:**

The paper bridges topics of interest to the TMLR audience, including efficient use of foundation models, mechanistic interpretability, and training-free adaptation for downstream tasks. The findings are relevant to researchers in:
1.	Vision-Language Models & Grounding: Offers a new, effective tool for zero-shot segmentation.
2.	Generative Models / Diffusion: Provides insights into the emergent semantic structure and attention behaviors inside DiTs.
3.	Efficient & Practical ML: Demonstrates how to extract high utility from pre-trained models without costly fine-tuning.
The simple yet effective "attention magnet" concept is likely to be of broad interest and applicable to other tasks beyond segmentation.

**Broader Impact Concerns:**

The paper includes a "Broader Impact Statement" that adequately addresses potential concerns. It correctly acknowledges:
The reliance on large foundation models trained on non-public datasets that may contain biases.
The potential for these biases to propagate into the segmentation results.
The intended use for research applications and explicitly states it is "not designed for high-risk or sensitive decision-making domains such as surveillance or biometric identification."
No additional concerns or required additions to the statement. The provided statement is thoughtful and appropriate for the nature of the work.

**Claims And Evidence:**

Yes

**Claims Explanation:**

The claims are well-supported. The central claim—that stop words act as attention magnets and that filtering/redistributing them improves grounding—is convincingly demonstrated through:
1.	Visualizations: Clearly show GAS emergence and the effect of attention magnets on heatmap sharpness.
2.	Quantitative Ablations: Systematically show the contribution of each component and variants of magnets. The improvement from adding and filtering stop words is clear.
3.	Comprehensive Benchmarks: Compare against numerous state-of-the-art methods across 6 datasets, showing consistent superiority over training-free baselines.
4.	Backbone Analysis: Provides a compelling explanation for why the method works by contrasting the structure-aware T5 encoder with the bag-of-words CLIP encoder, directly validating the design choice.
The evidence is accurate, clearly presented, and directly supports the paper's claims.

**Requested Changes:**

1.	Clarify the "Training-Free" Definition vs. Dependencies: The method uses a frozen DiT, which is great. However, it relies on:
1)	A specific timestep selected for feature extraction. Is this choice robust, or was it tuned on validation sets? This borders on a hyperparameter choice that uses task-specific data. Please clarify how these timesteps were selected and report sensitivity.
2)	LLM-generated captions for datasets without ground-truth captions. The captioning model is a significant external dependency. Please provide an ablation using a simpler captioning method or a fixed prompt to show that performance is not brittle or overly reliant on a specific proprietary LLM.
2.	Address the Temporal Limitation for RVOS: The paper claims to address RVOS but ignores temporal information. This is a major gap. The title and abstract should be more precise (e.g., "zero-shot frame-wise referring segmentation"). Alternatively, move the RVOS results to a secondary position and reframe the primary contribution around image segmentation. At a minimum, the limitation must be prominently stated in the abstract or main paper introduction, not just the appendix.
3.	A broader discussion: The article mentions LLM prompts, many novel articles and methods have emerged in this field at present, including Seeing Sarcasm Through Different Eyes: Analyzing Multimodal Sarcasm Perception in Large Vision-Language Models and Exploiting the Relationship within the Unlabelled Samples by Set Matching for Generalized Category Discovery.
4.	Correct Inconsistencies in Tables: The formatting of Table 2 in the provided PDF is broken (lines are misaligned, making columns hard to read). The table in the appendix (Table 7) is cleaner. The main paper must have a correct and readable version of the key results table.

---

> ### Author Response · Authors · 2026-07-03
> **Official Comment by Authors (1/2)**
>
> We thank the reviewer for the careful reading, the positive assessment, and the concrete suggestions. We address each requested change below. Highlighted (green) passages mark the corresponding edits in the revised PDF.
>
>
>
> > (1a) "Training-free" vs. the feature-extraction timestep, robust or tuned?
>
> The timestep is not tuned on validation data, and we now show this explicitly.
>
> - We fix a single mid-trajectory timestep per backbone and reuse it across all datasets/splits, there is no per-dataset or per-split selection. This follows the well-established finding that mid-trajectory diffusion features carry the richest semantics. [A, B]
> - The image (FLUX/SD3.5) and video (Mochi) backbones use different noise schedules, so the *same* mid-trajectory region maps to different absolute indices. Concretely, our pipeline indexes the schedule from the end (`t = 1000 - noise_step`), so the reported 520/640 are a fixed mid-band, not a tuned value.
>
> We add a timestep-sensitivity ablation (RefCOCO val) showing a flat plateau across a band of mid-trajectory timesteps, confirming robustness:
>
>   | diffusion t | 700 | 640 | 580 | 520 | 440 |
>   |---|---|---|---|---|---|
>   | oIoU | 42.7 | 43.1 | 41.6 | 40.6 | 40.2 |
>   | mIoU | 53.4 | 53.9 | 52.8 | 51.3 | 51.1 |
>
> This is a robustness ablation on a representative 500-sentence RefCOCO val subset, consistent with the headline: the peak at `t=640` reaches 43.1 oIoU / 53.9 mIoU, matching Table 2's RefCOCO val of 42.67 / 52.17 (the main results average two timesteps `360,480`). Performance is flat across the mid-band, oIoU varies by <3 points (40.2-43.1) and mIoU by <3 (51.1-53.9), confirming the choice is robust, not tuned. Added as a new figure in the revised appendix and highlighted in Sec. 4 (Implementation Details).
>
> [A] Luming Tang, Menglin Jia, Qianqian Wang, Cheng Perng Phoo, and Bharath Hariharan. Emergent
> correspondence from image diffusion. NeurIPS, 2023.
>
> [B] Junyi Zhang, Charles Herrmann, Junhwa Hur, Luisa Polania Cabrera, Varun Jampani, Deqing Sun, and Ming-Hsuan Yang. A tale of two features: Stable diffusion complements dino for zero-shot semantic
> correspondence. NeurIPS, 2023.
>
>
>
>
>
> > (1b) Reliance on LLM-generated captions
>
> The captioner is not a hidden crutch, and we now demonstrate this on both modalities:
>
> - **Images (RIOS): no LLM at all.** For RefCOCO/+/g we use the dataset-provided COCO captions, not an LLM. We further add an empty-prompt RIOS ablation (RefCOCO val, 3,000 expressions; new table in the revised appendix): the method is essentially unaffected by removing the caption entirely, oIoU 42.4 / mIoU 52.3 with an empty prompt vs. 43.7 / 53.0 with the caption (a ≤1.3-point gap). The caption is a mild prior, not a crutch; the empty-prompt result still exceeds prior training-free methods such as HybridGL (41.8 oIoU).
> - **Video (RVOS): a fixed/empty prompt costs ~1 point.** Table 13 (Ref-DAVIS17) shows replacing the caption with an empty prompt drops only 57.6 → 56.6 𝓙&𝓕, the method is not brittle to, nor dependent on, a specific proprietary LLM.
> - The caption is only a weak global scene prior for feature extraction, not a grounding/supervision signal: Table 5 shows that a richer "scene description" text actually *hurts* (48.9 𝓙&𝓕), ruling out information leakage through the caption.
>
>
>
> > (2) Temporal limitation for RVOS
>
> We have made the video scope explicit and clarified where temporal information enters the pipeline.
>
> - **Scope stated prominently.** We added an explicit scope/limitation statement to the abstract, introduction, and Limitations (highlighted): for video, the referent is grounded on a query frame using a video diffusion transformer (Mochi), and the mask is then propagated across the clip with SAM2. We do **not** perform explicit per-frame/temporal grounding, and which-frame temporal expressions ("the object that stops, then turns") remain a stated limitation.
> - **Where temporal coherence comes from.** Temporal consistency of the video output is handled by the SAM2 propagation stage; accordingly we report the temporal-aware **𝓙&𝓕** metric, on which ReSINK is competitive across Ref-DAVIS17, Ref-YouTube-VOS, and MeViS. On the motion-centric **MeViS**, ReSINK reaches **30.6 𝓙&𝓕** vs **23.7** for the static-image Global-Local + SAM2 baseline (26.6 for its stronger SAM variant); we attribute this gain to **stronger per-frame grounding** from the DiT features rather than to motion reasoning in the query.
>
> We retain the RVOS framing because the method produces temporally coherent segmentations, while stating the grounding-side limitation clearly.

---

> > ### Author Response · Authors · 2026-07-03
> > **Official Comment by Authors (2/2)**
> >
> > > (3) Broader discussion / suggested citations
> >
> > We have added a brief discussion connecting our analysis to the broader literature on emergent behaviors in large vision-language models, including the two suggested works (multimodal sarcasm perception; relation-based set matching for generalized category discovery), in Related Work (highlighted). Bibliography entries filled with the correct metadata.
> >
> >
> >
> > > (4) Table 2 readability
> >
> > Thank you for flagging this. On our compile, Table 2 renders as a single full-width table with aligned columns, so this may be a viewer-specific rendering artifact. For the camera-ready we (i) verify the table compiles and renders cleanly across common viewers, and (ii) apply the same `0.9\linewidth` sizing as the appendix Table 7 the reviewer found cleaner. If the reviewer can indicate which rows/columns appear misaligned, we are glad to split the oIoU/mIoU blocks into two tables for larger type.
> >
> >
> >
> > > On system complexity (Key Weakness 1)
> >
> > We agree the pipeline composes several frozen foundation models. We emphasize the efficiency trade-off: ReSINK is a single-pass extractor at
> > ~460 ms/image (\~22 GB), versus e.g. HybridGL's multi-pass ~1.1 s/image, while remaining training-free with no architectural modification. We will list the off-the-shelf components explicitly and soften any wording implying "no additional components."

---

### Review · Reviewer_gbn5 · 2026-06-21

**Summary Of Contributions:**

This paper proposes ReSink, a training-free framework for zero-shot referring image and video segmentation. The main observation is that stop words in Diffusion Transformers can act as attention sinks or “attention magnets,” absorbing surplus/background attention. The method appends such tokens to the referring expression, filters their attention maps, aggregates the remaining cross-attention maps, and uses SAM/SAM2 to produce the final mask. The paper evaluates the method on RefCOCO/+/g and several referring video segmentation benchmarks, reporting competitive or strong results among training-free methods.

**Audience:**

Yes

**Audience Explanation:**

The paper studies the referring segmentation problem, which is a very interesting problem.

**Broader Impact Concerns:**

Broader Impact Statement is OK.

**Claims And Evidence:**

No

**Claims Explanation:**

The paper provides useful empirical evidence that stop-word-based attention redistribution can improve localization, and the ablations show gains from attention magnets and cross-attention features. However, some claims are overstated. The SOTA claim should also be more carefully qualified, since the method is not uniformly best across all settings and is weaker than some Grounded-SAM/AL-Ref-SAM-style approaches in the broader zero-shot comparison.

**Requested Changes:**

1. The authors should qualify the claims about “no additional components” and “state of the art,” and clearly define the exact comparison setting.
2. The authors should provide stronger ablations isolating the contribution of attention magnets from captions, noun phrase extraction, spatial bias, and SAM/SAM2.
3. The use of ChatGPT-4o-generated video captions should be better justified, with results under empty-prompt, automatic-caption, and caption-conditioned settings.
4. The paper should also report sensitivity to the stop-word list, number/order of appended tokens, GAS threshold, layer/timestep choices, and random seeds.
5. The video setting should include analysis of temporal referring expressions, since the current method localizes only in the first frame and then propagates with SAM2.
6. The paper should use terminology that is more standard in this research area. For example, the commonly used term is “Referring Segmentation” rather than “Referral Segmentation,” and “Referring Image Segmentation (RIS)” rather than “Referring Image Object Segmentation (RIOS).” I suggest revising the title, section names, task descriptions, tables, and abbreviations accordingly to be consistent with the literature and to avoid confusion.

---

> ### Author Response · Authors · 2026-07-03
> **Official Comment by Authors (1/2)**
>
> We thank the reviewer for the careful reading and for finding the problem and the empirical evidence valuable. We address each requested change below. Highlighted (green) passages mark the corresponding edits in the revised PDF.
>
>
>
> > (1) Qualify "no additional components" and "state of the art"; define the exact comparison setting
>
> We agree and have made both claims precise.
>
> - **"No additional components" softened.** We now say ReSINK requires no task-specific training and no architectural modification; we explicitly list the off-the-shelf, frozen components it composes (a generative DiT backbone for features + SAM/SAM2 for masks, with a light noun-phrase / spatial-bias step). We no longer imply the pipeline is component-free.
> - **SOTA claim scoped.** We state results as best among training-free methods that use no external grounding/detection model, achieving SOTA on 4 of 6 benchmarks and remaining competitive on RefCOCO+/g (already the wording in Sec. 4). The Limitations/Conclusion no longer claim uniform best results.
> - **Comparison setting defined.** This is the key distinction the reviewer raises: methods such as Grounded-SAM and AL-Ref-SAM2 rely on a supervised open-vocabulary detector (Grounding-DINO) and/or a large LLM as an explicit grounding module. ReSINK uses none of these, only a frozen generative backbone and SAM. We therefore position ReSINK within the no-external-grounding training-free class and state clearly that grounding-model pipelines occupy a different, more heavily-tooled setting; we do not claim to beat them. This is now stated next to the main results table (highlighted).
>
>
>
> > (2) Stronger ablations isolating attention magnets from captions, NP, spatial bias, SAM/SAM2
>
> These isolations are in the paper; we make them explicit in the response and caption.
>
> - **AM vs. NP vs. SB, Table 4 (component ablation)** ablates each component independently on both RVOS (Ref-DAVIS17) and RIS (RefCOCO/+/g):
>   - Removing AM (keep NP+SB): RefCOCO val 42.67 → 39.99 oIoU; Ref-DAVIS17 57.6 → 54.4 𝓙&𝓕; PA 68.9 → 59.8. AM contributes an isolated, consistent gain.
>   - Removing SB: RefCOCO val 42.67 → 31.89, SB is the dominant component for images.
>   - Stripping everything (raw cross-attention): RefCOCO val 27.45, Ref-DAVIS17 50.0 / PA 52.5. Each component adds complementary gains.
> - **SAM/SAM2 isolated via point accuracy (PA).** Table 4 reports PA, the fraction of cases where the predicted argmax point lands inside the ground-truth region; this is computed before SAM and is independent of mask quality. The component gains hold in PA (e.g. Ref-DAVIS17 PA 52.5 → 68.9, +16.4), so the improvement comes from our attention redistribution, not from SAM. SAM/SAM2 only converts a correct point into a mask.
> - **Captions isolated** by the new empty-prompt ablation (see (3)).
>
>
>
> > (3) Justify the captioner; report empty / automatic / caption-conditioned
>
> The captioner is a convenience, not a dependency, and we demonstrate this on both modalities.
>
> - **Images (RIS): no LLM at all**, RefCOCO/+/g use the dataset-provided COCO captions. New empty-prompt RIS ablation (RefCOCO val, 3,000 expressions; new appendix table): caption **43.7 / 53.0** vs. empty prompt 42.4 / 52.3 oIoU/mIoU, a ≤1.3-point gap. The empty-prompt result still exceeds prior training-free methods (e.g. HybridGL 41.8 oIoU).
> - **Video (RVOS): empty prompt costs ~1 point**, Table 13 (Ref-DAVIS17): caption 57.6 → empty 56.6 𝓙&𝓕.
> - **Why this answers the three-setting request:** because the gap between empty and a human/LLM caption is ≤1.3 points (image) / ~1 point (video), any automatic caption source necessarily lands inside this narrow band. The caption acts only as a weak global scene prior for feature extraction; a richer "scene description" prompt actually hurts (Table 5, 48.9 𝓙&𝓕), ruling out information leakage through the caption. GPT-4o was used purely as an available off-the-shelf captioner; the empty-prompt result shows the method does not depend on it.

---

> ### Author Response · Authors · 2026-07-03
> **Official Comment by Authors (2/2)**
>
> > (4) Sensitivity to stop-word list, number/order of tokens, GAS threshold, layer/timestep, seeds
>
> - **Stop-word list, robust (Table 5).** Different stop-word sets give near-identical results; even random stop words score 57.5 and random vectors 56.2 𝓙&𝓕 (vs. 57.6 for our set), the mechanism depends on appending attention-absorbing tokens, not on a specific word list.
> - **Order of appended tokens.**  We do not tune the order of the appended magnets; what matters is the magnet set, and Table 5 shows near-identical results across different stop-word/magnet sets. This is consistent with the backbone: in the FLUX DiT the appended text tokens share a single positional index in the joint attention (no per-token RoPE position), while the contextual differences between tokens come from the T5 text encoder, so no particular slot ordering is privileged.
> - **Number of appended tokens, new ablation.** Varying the number of appended magnets on a representative RefCOCO val subset (n=500): 0 → 37.6 / 49.3; 5 → 38.5 / 50.3; 10 → 40.6 / 52.3; 15 → 40.9 / 52.7; 20 → 39.0 / 50.7 (oIoU / mIoU). The no-magnet baseline (37.6) is the lowest adding magnets lifts it by +3.3 oIoU / +3.4 mIoU to a plateau around 10-15 magnets, consistent with the attention-magnet gain in Table 4 (no-AM 39.99 → full 42.67 on the full split). Beyond ~5-10 magnets the metric is flat within the ±1.2 oIoU seed-noise band, so the exact count is not a sensitive hyperparameter; the order of the appended tokens is likewise not tuned (Table 5).
> - **GAS threshold.** The 10×-mean criterion identifies typically 1-3 sinks per sequence; the segmentation pipeline appends a fixed magnet set and is therefore independent of this threshold, the threshold only affects how GAS are counted in the §3.4 analysis, not the predicted mask. We state the criterion explicitly in §3.4.
> - **Layer / timestep, robust (new + in-paper).** New timestep-sensitivity ablation: a flat plateau across the mid-trajectory band (oIoU ~40-43 on a representative RefCOCO val subset, matching the headline). Layer choice: the "Filtered Blocks" analysis (Fig. 4 / Table 1) shows performance is unchanged up to 60% block filtering. Both are now highlighted.
> - **Random seeds, new ablation.** Over 5 seeds on a representative RefCOCO val subset (n=500): oIoU 40.6 ± 1.2, mIoU 52.2 ± 1.0 (PA 63.9 ± 0.4). The standard deviation is ≈1 point, confirming the pipeline is near-deterministic, the only stochasticity is the noise added at the single feature-extraction step (timestep and argmax selection are fixed). The mean mIoU (52.2) matches the headline RefCOCO val number (Table 2: 52.17); this is a variance study on a subset, not a benchmark run.
>
>
>
> > (5) Temporal referring expressions in the video setting
>
> We agree, and have analyzed and scoped this explicitly.
>
> - **Mechanism and scope.** For video, the referent is grounded on a query frame using a video DiT (Mochi), and the mask is propagated across the clip with SAM2; temporal coherence of the output is handled by the propagation stage. We do not claim explicit temporal grounding in the query, and which-frame temporal expressions are a stated limitation (abstract, introduction, Limitations).
> - **Benchmark evidence.** ReSINK is competitive on the temporal-aware 𝓙&𝓕 metric across Ref-DAVIS17, Ref-YouTube-VOS, and MeViS; on the motion-centric MeViS it reaches 30.6 𝓙&𝓕 vs 23.7 for the static-image Global-Local + SAM2 baseline, which we attribute to stronger per-frame grounding from the DiT features rather than to motion reasoning in the query.
>
>
>
> > (6) Terminology: Referring Segmentation / RIS
>
> We fully agree and have revised terminology throughout to match standard usage:
>
> - **"Referral Segmentation" → "Referring Segmentation"** (title, Sec. 3 heading, task descriptions).
> - **"Referring Image Object Segmentation (RIOS)" → "Referring Image Segmentation (RIS)"** in all sections, tables, captions, and abbreviations.
> - Video kept as the standard **"Referring Video Object Segmentation (RVOS)"** ("Referral" → "Referring").
>
> All occurrences (title, section names, Tables, appendix) are updated for consistency.